# Structural Determinants of the Prion Protein N-Terminus and Its Adducts with Copper Ions

**DOI:** 10.3390/ijms20010018

**Published:** 2018-12-20

**Authors:** Carolina Sánchez-López, Giulia Rossetti, Liliana Quintanar, Paolo Carloni

**Affiliations:** 1Department of Chemistry, Center for Research and Advanced Studies (Cinvestav), 07360 Mexico City, Mexico; magdacarolina29@hotmail.com; 2Institute of Neuroscience and Medicine (INM-9) and Institute for Advanced Simulation (IAS-5), Forschungszentrum Jülich, Wilhelm-Johnen-Strasse, 52425 Jülich, Germany; g.rossetti@fz-juelich.de; 3Jülich Supercomputing Center (JSC), Forschungszentrum Jülich, 52428 Jülich, Germany; 4Department of Oncology, Hematology and Stem Cell Transplantation, Faculty of Medicine, RWTH Aachen University, Pauwelsstraße 30, 52074 Aachen, Germany; 5Department of Physics and Department of Neurobiology, RWTH Aachen University, 52078 Aachen, Germany; 6Institute for Neuroscience and Medicine (INM)-11, Forschungszentrum Jülich, 52428 Jülich, Germany

**Keywords:** N-terminal prion protein, copper binding, prion disease mutations

## Abstract

The N-terminus of the prion protein is a large intrinsically disordered region encompassing approximately 125 amino acids. In this paper, we review its structural and functional properties, with a particular emphasis on its binding to copper ions. The latter is exploited by the region’s conformational flexibility to yield a variety of biological functions. Disease-linked mutations and proteolytic processing of the protein can impact its copper-binding properties, with important structural and functional implications, both in health and disease progression.

## 1. Introduction

Prion diseases or transmissible spongiform encephalopathies (TSEs) are rare neurodegenerative diseases exhibiting symptoms of both cognitive and motor dysfunction, vacuolation of the grey matter in the human central nervous system, neuronal loss, and astrogliosis [1]. A crucial event for the diseases’ development is the misfolding of the extracellular, membrane-anchored human prion protein (HuPrP^C^) into the fibril-forming isoform called “scrapie” (HuPrP^Sc^), the major or only component of the infectious particle [2]. This eventually leads to protofibril and fibrillar structures. Accordingly, with the “Protein only hypothesis” by Nobel Laureate Prusiner [3], the feature to undergo induced or spontaneous misfolding depends basically on intrinsic features of the protein. These include the amino acid sequence [4,5] as well as secondary structure elements [6,7,8], the highly flexible amino terminal region of the protein [9], and posttranslational modification elements [10]. The propensity to form the scrapie form is modulated by a variety of external factors. These include pH [11,12,13], cofactors like metal ions [14,15], or the presence of proteins [16,17]. Pathogenic mutations (PM) in HuPrP^C^ are linked to the spontaneous generation of prion diseases [18,19,20,21].

HuPrP^C^ is ubiquitously expressed throughout the body. It is mostly found in the central nervous system. After being synthesized in the rough endoplasmic reticulum, it transits through the Golgi compartment, and it is released to the cell surface where it resides in lipid membrane domains [22]. Though its physiological role is still not clear, HuPrP^C^ might be involved in neuronal development, cell adhesion, apoptotic events, and cell signaling in the central nervous system. Moreover, HuPrP^C^ can interact with different neuronal proteins or proteins of the extracellular matrix, as well as with other binders including glycosaminoglycans, nucleic acids, and copper ions [23]. Hence, HuPrP^C^ has been also proposed as a copper sensing or transport protein [24].

The protein features two signal peptides (1–22 and 232–235, Figure 1), a folded globular domain (GD, residues 125–231), and a naturally unfolded N-terminal tail (N-term_HuPrP^C^, hereafter, residues 23–124), which is the focus of this review. The GD consists of two β-sheets (S1 and S2), three α-helices (H1, H2, and H3), one disulfide bond (SS) between cysteine residues 179 and 214, and two potential sites for N-linked glycosylation (green forks in Figure 1) at residues 181 and 197 [25]. H2 and H3 helices linked by the SS-bond constitute the H2 + H3 domain. A glycosylphosphatidylinositol anchor (GPI, in blue in Figure 1) is attached to the C-terminus, which is located on the outside cellular membrane.

The HuPrP^C^→HuPrP^Sc^ interconversion involves mostly the GD. It may entail increasingly β-stranded intermediate structures [33] (Figure 1C), leading to small aggregates, protofibrils, and finally ordered rigid fibrils [34,35,36,37,38]. Experimental structural information for these is lacking [34,35,36,37,38].

While the structure of the GD of HuPrP^C^ has been resolved experimentally, the intrinsically disordered nature of the N-term_HuPrP^C^ has represented a challenge for structural studies. In this paper, the structural properties of the N-term_HuPrP^C^ are discussed, with a focus on recent insights obtained from computational approaches and on the functional and disease-related implications of copper–N-term_HuPrP^C^ interactions.

## 2. The N-Term: Function and Structural Determinants

This naturally unfolded domain contains the major part of the so-called transmembrane domain (termed TM1, comprising roughly residues 112–135) and the preceding “stop transfer effector” (STE, a hydrophilic region containing roughly residues 104–111) [39,40] (Figure 1B). STE and TM1 act in concert to control the co-translational translocation at the endoplasmic reticulum (ER) during the biosynthesis of the protein [41,42].

N-term_HuPrP^C^ functions as a broad-spectrum molecular sensor [43]. Along with the highly homologous protein from mouse (N-term_MoPrP^C^, 93% sequence identity), it interacts with copper ions (see below) and sulphated glycosaminoglycans [44]. In addition, N-term_MoPrP^C^ interacts with vitronectin [45], the stress-inducible protein 1 (STI1) [46], amyloid-β (Aβ) multimers [47,48,49], lipoprotein receptor-related protein 1 (LRP1) [50], and the neural cell adhesion molecule (NCAM) [51].

Because experimental structural information on the full-length N-term_HuPrP^C^ is currently lacking, one has to resort to biocomputing-based predictions. Recently, some of us have used a combination of bioinformatics along with replica-exchange-based Monte Carlo simulation at room temperature, based on a simplified force field, to predict the conformational ensemble on the full-length N-term_MoPrP^C^ [31,52].

This is expected to be quite similar to that from *Homo sapiens*, given the extremely high sequence identity (93%) with N-term_HuPrP^C^ [31,52]. Monte Carlo simulations suggest that the N-term_MoPrP^C^ consists of several regions characterized by different secondary structure elements, consistently with biophysical data [53,54,55,56,57]. Specifically, it contains 19 ± 8% α-helix, 8 ± 5% β-sheet, 7 ± 3% β-bridge, 27 ± 5% β-turn, 12 ± 4% bend, 4 ± 3% 3_10_-helix, and 1 ± 1% π-helix. The secondary structure elements are distributed among the N-term in a highly heterogeneous manner (Figure 2A): residues 23-30 are mainly coil/β-turn/bend; residues 31-50 are mainly β-turn/coil/bend/β-bridge; and residues 59-90 form four sequential octarepeat (OR) peptides, with sequence PHGGGWGQ, and are mainly β-turn/coil/bend/β-sheet conformations. In particular, the loop/β-turn conformations in the OR region resemble (backbone RMSD < 2.5 Å) those identified by NMR [57]; residues 89-98 are mainly coil/β-turn/bend/β-sheet; residues 99-117 feature the highest content α-helix of N-term_MoPrP^C^ regions; and residues 118-125 display a comparable percentage of α-helix and β-turn. Residues 105-125, the “amyloidogenic region”, feature transient helical structures (the last eight residues also have a comparable content of beta turn). This is consistent with circular dichroism (CD), nuclear magnetic resonance (NMR), and Fourier transform infrared (FTIR) studies on HuPrP^C^ fragments [54,55,56] (Figure 2). The same simulation procedure can be carried out for the known disease-linked mutations (Figure 2).

While many PMs in the GD are known to modify significantly the folded structure and to increase its flexibility [58,59,60,61], our Monte Carlo calculations suggest that those in the N-term do not impact significantly the global structural properties of the N-term. This finding is consistent with experimental findings showing that PMs in N-term_HuPrP^C^ do not affect the thermostability or misfolding kinetics of the protein [58,62,63,64]. On the contrary, our Monte Carlo simulations show that the PMs at the N-term modify local features at the binding sites for known cellular partners, as well as of interdomain interactions. This points to an interference of the PMs with the related physiological functions.

The major differences in the presence of the PMs were observed in the residues binding Cu^2+^ and sulphated GAG (i.e., the OR region and the H110 Cu^2+^-binding site mouse sequence, H111 in the human sequence). In addition, the PMs affect the SS and the flexibility and increase the hydrophobicity of STE/TM1. The latter contains the putative binding sites for in vivo binding partner proteins such as vitronectin [45] and STI1 [46]. This might affect the biological function of these interactions, which involves the signaling for axonal growth [45] and that for neuroprotection [46], respectively.

The PM Q52P in the OR region, interestingly, affects the flexibility of STE/TM1, while the other six PMs in STE/TM1 also alter the intra-molecular contacts in the OR region suggesting a role played by PMs in altering transient interdomain interactions between the OR region and STE/TM1. Recent studies suggest that N-Term and GD interactions might also serve to regulate the activity and/or toxicity of the PrP^C^ N-term [65]. Unfortunately, in the reported Monte Carlo study [52], the GD was not taken into account.

The altered local features in STE/TM1 might also impact the interactions of the protein with trans-acting factors in the cytosol and in the ER membrane [66]. This result is consistent with the in vitro data that PMs P101L, P104L, and A116V increase the interactions between MoPrP^C^ STE/TM1 and a membrane mimetic at pH 7 [67].

## 3. Copper Binding

Copper ions bind to the N-term of HuPrP^C^ in vivo [24]. Since the protein is anchored to the neuronal membrane, facing the extracellular space, it is exposed to fluctuations in Cu^2+^ ion concentrations, that can reach 100 µM during synaptic transmission [68]. This represents orders of magnitude larger than the experimentally measured range of binding affinities for Cu(II) ions at the N-term (nM to µM) [69]. Thus, it is plausible that the protein responds to Cu(II) ion concentration changes at the synapse. On the other hand, upon endocytosis, HuPrP^C^ is exposed to the intracellular reducing environment, where the interaction of its N-term with Cu^+^ ions would also be relevant. Two main functions of copper binding to the N-term have been proposed so far: (i) stimulation of HuPrP^C^ endocytosis [70,71,72,73] and (ii) copper sensing associated to cell signaling. Copper-induced endocytosis of HuPrP^C^ requires its N-term terminus, specifically the octarepeat region, and it might involve conformational alterations of the N-term with the subsequent delivery of copper ions to endosomes. This has led to a proposed role for HuPrP^C^ in copper transport. However, it is unlikely that HuPrP^C^ delivers copper efficiently into the cytosol, since high concentrations are needed for copper-induced endocytosis (150–300 µM) [71,73]. On the other hand, HuPrP^C^ can interact with the human N-methyl-D-aspartate receptors (HuNMDAR) and alpha-amino-3-hydroxy-5-methyl-4-isoxazolepropionic acid receptors (HuAMPAR) involved in synaptic transmission, while both receptors are regulated by Cu ions [74,75,76]. In the case of HuNMDAR activity, copper binding to HuPrP^C^ is necessary to regulate the activity of this receptor [75]. Indeed, HuPrP^C^ and Cu^2+^ are required to inhibit HuNMDAR activity through a mechanism that involves post-translational S-nitrosylation of cysteine residues in HuNMDAR [77]. Overall, these important findings underscore a role for Cu-HuPrP^C^ interactions in neuroprotective mechanisms, which could be disrupted by other Cu-binding proteins or peptides at the synapse. For instance, human amyloid-β (Aβ) neurotoxicity has recently been linked to its ability to compete for Cu^2+^ ions with HuPrP^C^, thereby interfering with the modulation of HuNMDAR activity [75].

The N-term region of HuPrP^C^ contains six His residues that may serve as anchoring sites for Cu^2+^ ions [78]. The ion binds to different sites within the protein [79,80,81,82], which are conserved in mammalian species [83], a fact that underscores its physiological relevance. Four of them are located in the OR region, spanning residues 60-91 with four repeats of the highly conserved octapeptide PHGGGWGQ (Figure 3). Beyond the OR region, two additional His residues, 96 and 111, also act as copper-binding sites in the 92-115 region. Studies on synthetic peptide fragments have suggested that metal coordination modes depend on copper concentration [69], as well as the relative copper:protein ratio and proton concentration [79,80]. At physiological pH, three distinct Cu^2+^ coordination modes have been identified by electron paramagnetic resonance (EPR) [84]. At low Cu:protein ratios, three or four His residues can chelate one metal ion, leading to a multiple histidine Cu-binding mode, named Component 3 (Figure 3). At higher Cu:protein ratios, a species with two His ligands forming a 2N2O equatorial coordination mode is observed (Component 2 in Figure 3). When enough Cu^2+^ is provided to reach a 1:1 ratio for each octapeptide fragment, a species with a 3N1O equatorial coordination mode is formed, named Component 1, where the coordinating residues are as follows: one His imidazole ligand, two deprotonated backbone amide groups, and a carbonyl group from the glycine residues that follow the anchoring His in the sequence (Figure 3) [78,85]. X-ray crystallographic studies of the Cu^2+^ complex with one octapeptide PHGGGWGQ fragment also revealed the participation of a water molecule as an axial ligand, stabilized by hydrogen bonding to the Trp residue [86]. This is the only Cu-binding site fragment that has been characterized so far by X-ray crystallography.

The K_d_ for the low occupancy multiple-His coordination mode (Component 3) is 0.12 nM, whereas it ranges from 7 to 10 µM for the high-occupancy 3N1O mode (Component 1) [69]. Hence, Cu binding to the OR region displays a negative cooperativity. This is consistent with the formation of intermediate species such as Component 2. Electrochemical studies have determined that the high-occupancy Component 1 species is capable of reducing dioxygen to produce low levels of hydrogen peroxide that may be relevant for cell signaling, whereas the low-occupancy multiple-His mode cannot activate dioxygen at all [87,88].

Outside the OR region, His 96 and 111 act as anchoring sites for Cu^2+^ ions, constituting the closest Cu-binding sites to the amyloidogenic region of HuPrP^C^ [81,89,90]. An X-ray absorption spectroscopy (XAS) study of a HuPrP (90–231) construct of the protein showed that at low pH Cu ions can coordinate to both His residues in the non-OR region, while at physiological pH only the His111 site is populated [91]. The peptide fragments mostly used to characterize the individual non-OR Cu-binding sites are HuPrP (92-96) and HuPrP (106-115) with sequences GGGTH and KTNMKHMAGA, respectively [85,92]. EPR and other spectroscopic studies have determined that Cu^2+^ coordination is highly pH-dependent in these sites, yielding two different equatorial coordination modes at physiological pH, 3N1O and 4N, related by a pKa value near 7.5 (7.8 for the His 96 site and 7.5 for the His 111 site) [93,94,95,96]. In both sites, Cu^2+^ coordination in the 3N1O mode involves the His imidazole group, two deprotonated amide nitrogens, and a carbonyl group from the backbone amide bonds that precede the His residue in the sequence, while a third deprotonated amide group replaces the carbonyl moiety in the 4N equatorial mode (Figure 3). Although the equatorial coordination modes of Cu^2+^ bound to these sites are identical, the presence of two Met residues in the His 111 site provides it with distinct properties, particularly in terms of relative binding affinity, and redox properties. The thioether groups of Met 109 and Met 112 can participate in Cu^1+^ coordination to the PrP(106-115) fragment, as demonstrated by XAS studies, yielding coordination modes where the His111 imidazole ring, the two Met residues, and a backbone carbonyl group stabilize a tetra-coordinated Cu^1+^ species at physiological pH, where Met109 plays a more important role in metal coordination, as compared to Met112 [97]. Anchoring of Cu^1+^ ions by Met residues persists even at low pH values (<5), as those found in endosomes. Thus, the MKHM motif and Cu coordination features of the His111 site assure that the metal ion would still be bound to the protein, even under decreased pH and reducing conditions, such as those encountered upon endocytosis. Additionally, the capability of the His111 site to stabilize both Cu^2+^ and Cu^1+^ makes it a unique site in the N-term region of HuPrP^C^ that may support redox activity to activate dioxygen.

The relative binding affinity of Cu^2+^ for the non-OR sites has also been studied [98,99,100,101,102]. While a slight preference for Cu^2+^ binding to His111 over His96 has been observed spectroscopically and ascribed to the Met residues nearby, the two sites get loaded simultaneously with K_d_ values in the range of 0.4 to 0.7 µM at physiological pH [102]. Given the Cu-binding affinity features of the OR region, this implies that upon increasing Cu^2+^ levels, the multiple His species (Component 3) would first form, followed by the population of the non-octarepeat His96/His111 sites, before the OR region is fully loaded to yield the high occupancy mode (Component 1). Overall, the Cu(II) coordination and binding features of the N-term region provide HuPrP^C^ with the ability to respond to a wide range of Cu concentrations, adopting different metal coordination modes, which in turn may impose different conformations to this unstructured region of the protein.

The conformational flexibility of the N-term_HuPrP^C^ and the presence of several His residues as Cu anchoring sites provide a platform to accommodate different Cu^2+^ coordination modes as a function of relative metal:protein concentrations. Unlike the static (entatic) Cu active sites of cuproenzymes, where the protein structure imposes restrictions on the metal coordination and geometry, the preferred coordination modes at each Cu-binding site in the flexible N-term domain of HuPrP^C^ are dictated by the geometric and electronic preferences of the metal ion, which can actually impose metal-induced conformations with potentially different functional implications or a propensity to aggregate. For example, the participation of deprotonated backbone amides in Cu^2+^ coordination, as in the high-occupancy (Component 1) mode of the OR region, inevitably imposes a certain turn in the backbone chain, which is known to yield more compact conformations [81]. Indeed, full loading of Cu^2+^ ions into the OR region yields a conformation where the average Cu–Cu distance is 4 to 7 Å, as determined by EPR [84]. Given the relatively high Cu concentrations needed for endocytosis of HuPrP^C^ and the Cu-binding affinity features of the OR region, this high-occupancy conformation of the OR region is likely involved in the mechanism of endocytosis.

Recent studies have revealed metal-induced interactions between the N-term and C-term regions of MoPrP^C^ [103,104,105]. An NMR study suggested interactions of the region 90-120, containing His96 and His111 Cu-binding sites, with residues in the vicinity of helix-1 (specifically 144-147), while the Cu-loaded OR region may also interact with helix-2, involving residues 174-185 [105]. The latter was confirmed by an elegant NMR and site-directed spin labeling EPR study that provided detailed structural information on how Cu binding at the low occupancy multiple His site (Component 3) in the OR region promotes electrostatic interactions with a highly conserved negatively charged region at helices 2 and 3 of the globular protein [103]. The C-term region of the protein engaged in the interaction with Cu-loaded OR overlaps with the region where neurotoxic PrP antibodies bind, and it also involves acidic residues associated with disease-related mutations, such as E200K. These observations underscore the important role of Cu^2+^ loading into the low-occupancy multiple-His coordination mode in promoting electrostatic interactions between the Cu-bound N-terminal region and the helical C-terminal domain, a stabilizing interaction that is considered to be regulatory for prion conversion [103,105].

On the other hand, in the non-OR region, the two His coordination modes identified at low pH for the PrP(90-231) construct were also found to induce stabilizing interactions with the globular C-terminal domain, whereas Cu^2+^ binding solely to the His111 site induced local beta-sheet structure [91]. Consistently, copper binding to the non-OR sites in the amyloidogenic fragment 90-126 induces a β-sheet-like transition [106]. These observations underscore the important role that Cu binding to the non-OR region may play in amyloid aggregation and prion conversion.

Cu^2+^-PrP^C^ interactions and their perturbation by disease-related mutations have been suggested to play a role for Hu/Mo PrP^C^ aggregation and prion disease progression [107]. Specifically, the GSS-linked Q211P PM [60] (Q212P in Hu numbering) in the HuPrP^C^ GD can influence the Cu^2+^ binding coordination at H96 and H111 [108], implicating a role of abnormal Cu^2+^ binding in the pathology of PMs in HuPrP^C^. As discussed above, the multiple His Cu-binding modes induce stabilizing interactions of the N-term_HuPrP^C^ with the globular C-terminal domain, whereas Cu^2+^ binding solely to the His111 site induces local beta-sheet structure [91,103,105]. Thus, any perturbation of the local conformation around the Cu-binding sites may have an impact on the stability of the protein. Consistently, structural analysis by molecular simulations of the N-term_HuPrP^C^ indicates that some disease-linked mutations may affect the local conformation and intramolecular interactions around the Cu^2+^ binding sites, including His111 [31,32], providing a molecular basis to understand their impact on disease progression. Although further studies are needed to understand how Cu binding impacts the folding and conformation of the flexible N-term_HuPrP^C^, it is clear that the different Cu^2+^ coordination modes formed at the His anchoring sites can favor distinct metal-induced conformations, while disease-related mutations may also impact the conformation of the N-term region, its Cu-binding properties, and its interactions with the C-terminal globular region of HuPrP^C^.

Copper binding may also be affected by a specific post-translational modification, namely the proteolytic processing at specific sites of the N-term region [109]. This includes the following: (i) the β-cleavage of the OR region, leading to the N2 (23-89) and C2 (90-231) fragments [109]. This is induced by reactive oxygen species (ROS) produced in the presence of Cu^2+^ ions [110,111] (It can be also catalyzed by calpain and ADAM8—a member of the A Disintegrin And Metalloproteinase (ADAM) family of enzymes [112,113]). While the N2 fragment may be released, maintaining the Cu^2+^ binding properties of the OR sites as described above, the C2 fragment may remain anchored to the membrane, conserving the His96 and His111 sites, but with a free N-term group at residue 90 [109]. The free NH_2_ moiety is expected to change significantly the Cu^2+^ coordination features of these non-OR sites. (ii) The α-cleavage occurs at several sites in the region encompassing residues 109-120, and it is a common feature in a wide range of cell lines [114]. The process, performed by members of the ADAM enzymes [112], increases in the presence of Cu^2+^ ions [115]. This metal also may modulate the relative amount of α-cleavage at each site, possibly by inducing local conformational changes that impact how the protein docks into the protease active site [112,113]. The most described α-cleavage site is located between Lys110 and His111 in the human sequence, producing two fragments N1 (23-110) and C1 (111-231) [109]. The released N1 fragment may still contain the OR region and the His96 site; however, the His111 site may be significantly disturbed, as the cleavage occurs between residues that participate in Cu^2+^ coordination, leaving a His111 with a free NH_2_ terminal group at the membrane bound C1 fragment. A recent spectroscopic study determined the impact of α-cleavage processing on Cu^2+^ binding to His111, using a model peptide for the C1 fragment [116]. Indeed, in this fragment His111 and the free NH_2_ terminal group act as the main anchoring sites for Cu^2+^, resulting in coordination modes that are highly dependent on proton and copper concentrations, and are quite different from those characterized for the intact His111 site in the full protein (Figure 3). The Cu-binding affinity features and redox activity of this perturbed His111 site remain to be investigated. It is interesting to note that, while Cu ions can modulate the relative amount of α-cleavage at different sites of the 109-120 region of HuPrP^C^ [112], the resulting membrane-bound C1 fragments and their Cu-binding properties could in turn determine the metal-induced conformation of the N-term region and its ability to interact with important receptors, such as the HuNMDAR and HuAMPAR [113,117].

## 4. Conclusions

Recent advances in computational biophysics [31,32,52] have led, for the first time, to a description of the conformational ensemble on the full-length N-term MoPrP^C^, a fully disordered domain of 125 amino acids, with high similarity to the human domain. This has made it possible to probe the impact of disease-related mutations on the structural properties of this flexible region of the protein.

N-term_HuPrP^C^ binds copper ions in vivo [24]. It yields a diverse range of Cu coordination modes, each with distinct redox properties and binding affinity features. The Cu-binding properties of the N-term region provide HuPrP^C^ with the ability to respond to the wide range of Cu concentrations that the protein is exposed to at the synapse, adopting different metal-induced conformations, which in turn may have distinct functional implications. On the other hand, Cu^2+^-PrP^C^ interactions and their perturbation by disease-related mutations may play a role in protein aggregation and prion disease progression.

While the interplay between metal ion binding and conformational flexibility in the entire N-term remains to be understood, it is well established that copper displays site-specific effects on its folding, either by promoting stabilizing interactions or inducing conversion to beta-sheet folds. Conversely, molecular simulations suggest that some disease-related mutations may affect the local conformation around the Cu anchoring sites, thus affecting the Cu-binding properties of the N-term_HuPrP^C^ and the stability of the protein.

Combined computational and experimental studies on the structural impact of Cu^2+^ binding and disease-related mutations at the N-term_HuPrP^C^, such as those on copper(II)-alpha-synuclein—an intrinsically disordered protein undergoing fibril formation in Parkinson’s disease [118]—could advance dramatically our understanding of the functional role of Cu^2+^-PrP^C^ interactions in health and disease.

## Figures and Tables

**Figure 1 ijms-20-00018-f001:**
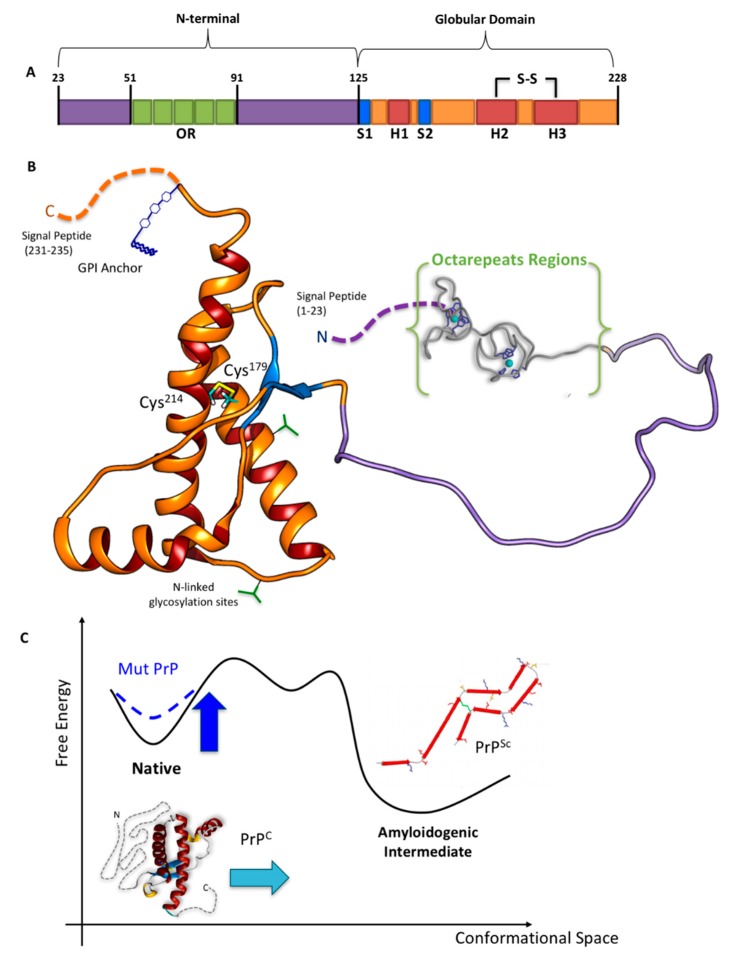
(**A**) Schematic and (**B**) tridimensional view of HuPrP^C^. (**C**) Qualitative scheme illustrating the Gibbs free energy change in the conversion from HuPrP^C^ (left) to HuPrP^Sc^ (right) [26]. The depicted amyloidogenic intermediate is the parallel, in-register β-structure model for the core of recombinant PrP90–231 amyloid fibrils formed in vitro [27], one of the models among others [28,29,30], whereas the native globular domain (GD) of the HuPrP^C^ is the nuclear magnetic resonance (NMR) structure by Zahn et al. [25]. Adapted from [31,32].

**Figure 2 ijms-20-00018-f002:**
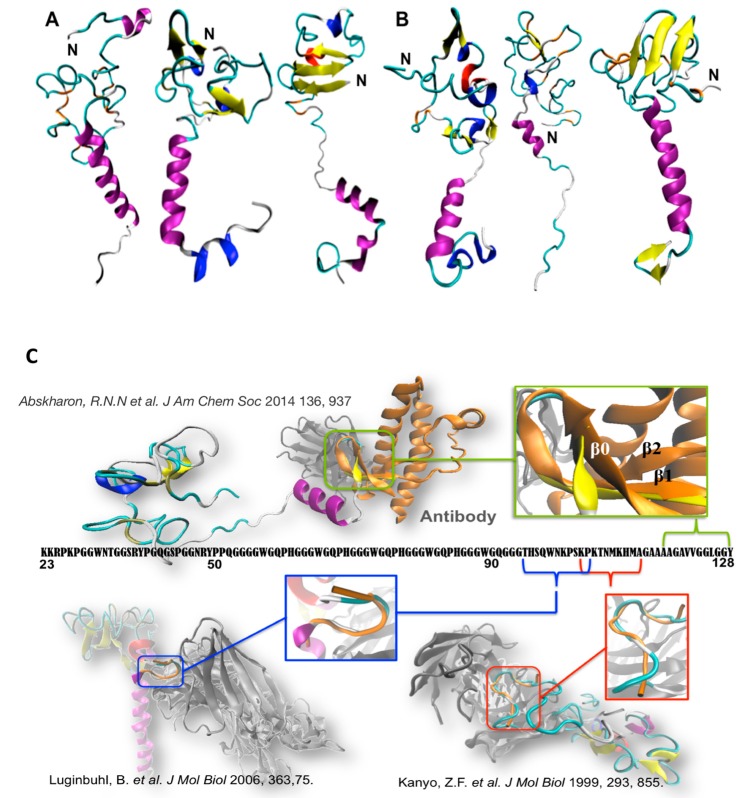
Selected conformations of (**A**) WT N-term_MoPrP^C^ and (**B**) one PM (N-term_MoPrP^C^_Q52P) emerging from molecular simulation [31,52]. These contain transient α-helix (in violet), β-sheet (yellow), β-bridge (orange), β-turn (cyan), 3_10_-helix (blue), and p-helix (red) elements. (**C**) Superimposition of our conformational ensemble (orange) with available fragments of N-term deposited structures. Readapted from [31,52].

**Figure 3 ijms-20-00018-f003:**
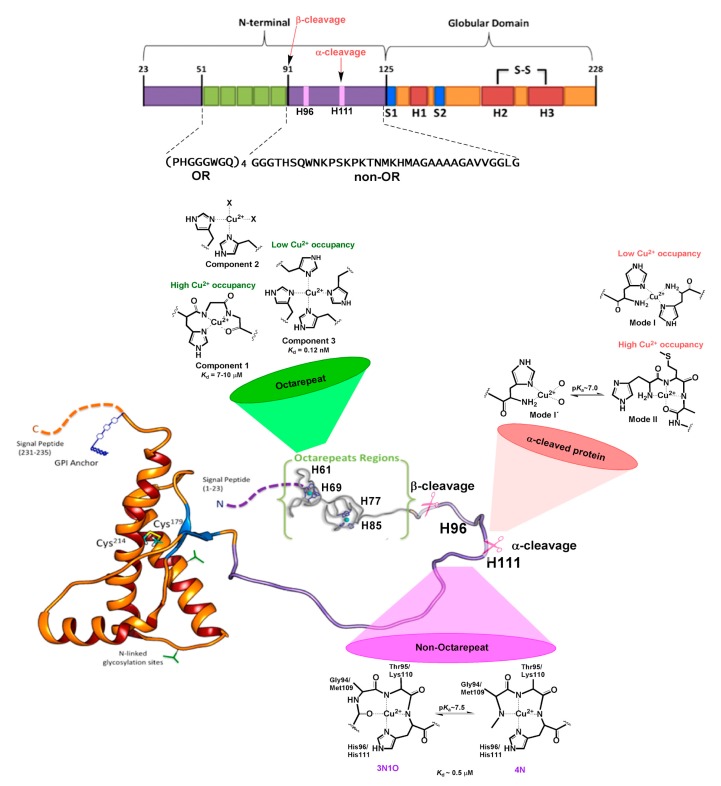
Cu coordination properties of the N-terminal region of human HuPrP^C^. The six His residues that act as anchoring sites for Cu ions are highlighted: His61, His69, His77, and His85 in the OR region, and His96 and His111 in the non-OR region. The models for the different Cu^2+^ coordination modes identified at each site at physiological pH are drawn. The impact of α-cleavage processing on the His111 binding site is also shown.

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
