# Peer review of "Structural Determinants of the Prion Protein N-Terminus and Its Adducts with Copper Ions"

_ijms, 2018, doi:10.3390/ijms20010018_

Round 1

Reviewer 1 Report

This manuscript by Sanchez-Lopez, et al. provides a thorough and up to date review of copper binding properties in the prion protein (PrP). The coverage of PrP structure, copper binding sites, binding affinities, the role of the protein's N-terminal domain in function, etc. is accurate and balanced. This work will give the reader a very helpful perspective on the current understanding of PrP biophysical chemistry and function. As such, I fully support publication. 

Some comments and recommendations:

1) In the Conclusions, the authors point to contributions of computer simulations towards the understanding of disease related mutations. This aspect of the review was not well developed earlier in the manuscript. Also, while the manuscript touches on the work of Evans et al. (ref 91), it overlooks what Evans et al. developed with respect to familial mutations disrupting PrP domain-domain interactions. This is reviewed in Evans et al. Progress in Molecular Biology and Translational Science Volume 150, 2017, Pages 35-56. Specifically, a significant number of familial mutations are at the domain-domain interface.

2) Line 6, page 2, "HuPrPC is ubiquitously expressed in the body." This sentence lacks specificity. I'd recommend "HuPrPC is ubiquitously expressed throughout the body."

3) Figure 1C  There is little experimental support for the combined beta-helix + alpha-helix structure of PrP-Sc shown in this Fig. The authors should consult the recent work of Wille and coworkers, and Surewicz and coworkers who show that beta-helical structure likely extends throughout the C-term domain.

4) Page 6, line 8  "The ion binds to different binding sites..."  Awkward wording. It would be sufficient to say "The ion binds to different sites..." or "The ion binds to different sites within the protein..."

Author Response

Responses to Reviewer 1

This manuscript by Sanchez-Lopez, et al. provides a thorough and up to date review of copper binding properties in the prion protein (PrP). The coverage of PrP structure, copper binding sites, binding affinities, the role of the protein's N-terminal domain in function, etc. is accurate and balanced. This work will give the reader a very helpful perspective on the current understanding of PrP biophysical chemistry and function. As such, I fully support publication.

RESPONSE: We acknowledge the referee for appreciating our efforts

Some comments and recommendations:

1) In the Conclusions, the authors point to contributions of computer simulations towards the understanding of disease related mutations. This aspect of the review was not well developed earlier in the manuscript. Also, while the manuscript touches on the work of Evans et al. (ref 91), it overlooks what Evans et al. developed with respect to familial mutations disrupting PrP domain-domain interactions. This is reviewed in Evans et al. Progress in Molecular Biology and Translational Science Volume 150, 2017, Pages 35-56. Specifically, a significant number of familial mutations are at the domain-domain interface.

RESPONSE: The referee is right! We have further improved the discussion of the disease-related mutation in the new version of the paper.

2) Line 6, page 2, "HuPrPC is ubiquitously expressed in the body." This sentence lacks specificity. I'd recommend "HuPrPC is ubiquitously expressed throughout the body."

RESPONSE: We have corrected the sentence.

3) Figure 1C  There is little experimental support for the combined beta-helix + alpha-helix structure of PrP-Sc shown in this Fig. The authors should consult the recent work of Wille and coworkers, and Surewicz and coworkers who show that beta-helical structure likely extends throughout the C-term domain.

RESPONSE: We thank the referee for his/her suggestion. We have changed figure 1C.

4) Page 6, line 8  "The ion binds to different binding sites..."  Awkward wording. It would be sufficient to say "The ion binds to different sites..." or "The ion binds to different sites within the protein..."

RESPONSE: We have corrected the sentence. 

Reviewer 2 Report

This review authored by Sánchez-López et al., is a detailed work covering the state of the art of the copper role in the prion protein folding and functions. It is well know that copper ions bind to five distinct binding sites in the N-terminal unfolded octarepeats and non-octarepeats regions and that the metal has structural effects on the plasticity of this domain. However, recent findings have highlighted that copper mediates functional interactions between the N- to C-terminal domains. This topic has not been addressed by authors and I encourage them to add a critical paragraph about this crucial part.

See, for instance, literature from Wells et al Biochem. J. 2006; Thakur et al JBC 2011; Younan et al, JMB 2011; Spevacek et al Structure 2013; to Evans et al Structure 2016 etc.

Author Response

Responses to Reviewer 2

This review authored by Sánchez-López et al., is a detailed work covering the state of the art of the copper role in the prion protein folding and functions. It is well know that copper ions bind to five distinct binding sites in the N-terminal unfolded octarepeats and non-octarepeats regions and that the metal has structural effects on the plasticity of this domain. However, recent findings have highlighted that copper mediates functional interactions between the N- to C-terminal domains. This topic has not been addressed by authors and I encourage them to add a critical paragraph about this crucial part.

See, for instance, literature from Wells et al Biochem. J. 2006; Thakur et al JBC 2011; Younan et al, JMB 2011; Spevacek et al Structure 2013; to Evans et al Structure 2016 etc.

RESPONSE: We thank the reviewer for the constructive comment. In the original version of the manuscript we had already cited Thakur eta al JBC 2011 and Evans et al Structure 2016. We have now expanded the discussion on the Cu-mediated functional interactions between N- and C- domains of PrP.